# Skin Cancer Detection Using Transfer Learning and Deep Attention Mechanisms

**DOI:** 10.3390/diagnostics15010099

**Published:** 2025-01-03

**Authors:** Areej Alotaibi, Duaa AlSaeed

**Affiliations:** College of Computer and Information Sciences, King Saud University, Riyadh 11451, Saudi Arabia; dalsaeed@ksu.edu.sa

**Keywords:** skin cancer, deep learning, transfer learning, attention mechanism, computer vision, pre-trained models, Xception, dermoscopic images, medical imaging

## Abstract

**Background/Objectives:** Early and accurate diagnosis of skin cancer improves survival rates; however, dermatologists often struggle with lesion detection due to similar pigmentation. Deep learning and transfer learning models have shown promise in diagnosing skin cancers through image processing. Integrating attention mechanisms (AMs) with deep learning has further enhanced the accuracy of medical image classification. While significant progress has been made, further research is needed to improve the detection accuracy. Previous studies have not explored the integration of attention mechanisms with the pre-trained Xception transfer learning model for binary classification of skin cancer. This study aims to investigate the impact of various attention mechanisms on the Xception model’s performance in detecting benign and malignant skin lesions. **Methods:** We conducted four experiments on the HAM10000 dataset. Three models integrated self-attention (SL), hard attention (HD), and soft attention (SF) mechanisms, while the fourth model used the standard Xception without attention mechanisms. Each mechanism analyzed features from the Xception model uniquely: self-attention examined the input relationships, hard-attention selected elements sparsely, and soft-attention distributed the focus probabilistically. **Results:** Integrating AMs into the Xception architecture effectively enhanced its performance. The accuracy of the Xception alone was 91.05%. With AMs, the accuracy increased to 94.11% using self-attention, 93.29% with soft attention, and 92.97% with hard attention. Moreover, the proposed models outperformed previous studies in terms of the recall metrics, which are crucial for medical investigations. **Conclusions:** These findings suggest that AMs can enhance performance in relation to complex medical imaging tasks, potentially supporting earlier diagnosis and improving treatment outcomes.

## 1. Introduction

Skin cancer is one of the most devastating cancers of the present decade, and it is the fifth most common form of cancer [1]. According to the World Health Organization (WHO), melanoma, its most aggressive form, affects over 132,000 new patients worldwide each year [2]. In the United States, skin cancers constitute 4% of all malignant neoplasms, with approximately 1% of all cancer-related deaths attributed to this disease [3]. Within our region, the latest Saudi Cancer Registry showed that skin cancer ranked as the ninth most common cancer among both sexes, representing 3.2% of all newly diagnosed cases in 2010 [4]. Despite this, there is a significant chance in terms of survival of skin cancer when it is caught at an early stage. According to the Centers for Disease Control and Prevention (CDC), the treatment for all kinds of skin cancer costs at least USD 8 billion per year [5].

Skin cancer develops when abnormal cells in the epidermis (the outermost layer of the skin) grow uncontrollably. The condition begins when DNA damage goes unrepaired, triggering mutations that cause skin cells to multiply rapidly and form malignant tumors. While skin cancer can occur anywhere on the body, it most commonly appears on sun-exposed areas like the head, face, lips, ears, neck, chest, and arms, as well as the legs in women.

The development of skin cancer involves multiple pathways and complex interactions among genetic, environmental, and behavioral factors. While UV radiation from sun exposure remains the primary environmental risk factor, causing DNA damage that can trigger carcinogenesis, other significant factors include genetic predisposition, immunosuppression, and exposure to carcinogenic chemicals [6].

Risk levels vary considerably across populations. Individuals with lighter skin tones, blonde or red hair, and a tendency toward sunburn face a higher risk. The lifetime risk of developing melanoma is 2.4% in Caucasians, 0.5% in Hispanics, and 0.1% in Black individuals [7]. Genetic factors also play a crucial role, with studies indicating that 7–15% of melanoma patients have a family member with a history of the disease [8].

Notably, skin cancer affects all ages and genders, but disparities exist. For example, the American Academy of Dermatology reports [9] that men, especially older men, are at higher risk. By age 65, men are twice as likely as women to develop melanoma, and by 80, three times as likely. Men also face a higher mortality risk from melanoma across all age groups.

Skin cancer includes two primary categories: malignant melanoma (MM) and non-melanoma skin cancer (NMSC), each exhibiting distinct clinical outcomes. According to the WHO, global cases of melanoma and NMSC are estimated to exceed 1.7 million new diagnoses in 2025 [10]. Malignant melanoma is the least common form of skin cancer, accounting for only 5% of cases, and early detection plays a crucial role in determining the patient’s prognosis [11]. While NMSC does not significantly contribute to the overall cancer mortality, its incidence increases with age. A study presented at the Conference (EADV) 2023 revealed that NMSC is responsible for more global deaths than melanoma skin cancer [12]. The two primary types of NMSC are basal cell carcinoma (BCC) and squamous cell carcinoma (SCC), with other types being less common. BCC accounts for approximately 80% to 85% of NMSC cases, while SCC comprises 15% to 20%. SCC demonstrates a higher propensity for metastasis compared to BCC [11]. The contrast in the propensity for metastasis between MM and NMSC positions MM as the primary contributor to mortality among individuals with skin cancer.

As mentioned, early diagnosis significantly increases the likelihood of recovery. Detecting skin cancer in its initial stages not only facilitates easier treatment but also enhances the overall prognosis. According to Cancer Research UK, the chances of successful treatment are higher when cancer is discovered before it has advanced or spread [13].

Dermatologists commonly diagnose skin cancer through conventional histopathology (biopsy), an invasive clinical method. This procedure involves removing a sample from a suspected skin lesion for examination to determine malignancy. While biopsies achieve nearly 100% diagnostic accuracy [14], they are costly, time-consuming, and painful for patients. In the Medicare population alone, over 8.2 million skin biopsies are performed annually to diagnose approximately 2 million skin cancers, leading to numerous unnecessary procedures, scarring, and high financial costs [15]. Additionally, the complete process—from biopsy and tissue processing to pathologist review and diagnostic assessment—can take anywhere from one day to several weeks, creating a significant delay between initial assessment and treatment [16].

As a result, noninvasive techniques like macroscopic and dermoscopic imaging have become essential for skin evaluation. While microscopic images from mobile cameras offer low resolution, dermoscopy provides high-resolution images that reveal deeper skin structures, improving diagnostic accuracy and potentially reducing mortality rates [17]. Expert dermatologists analyze these images through visual inspection, a process that requires skill, attention, and significant time [18]. As there are over 2000 dermatological diseases, similar-looking skin lesions from different conditions can complicate visual examinations and increase the risk of misdiagnosis [19].

Artificial intelligence (AI) and deep learning (DL) have revolutionized skin cancer diagnostics, offering significantly higher prediction accuracy compared to traditional visual examinations. The convolutional neural network (CNN), a prominent deep learning model, has shown exceptional performance in diagnosing skin lesions, as seen in studies [10,11], among others. However, deep learning models in medical image analysis face data scarcity challenges. Medical image acquisition and annotation are expensive and time-intensive processes. While data sharing could address this limitation, it raises ethical concerns and privacy issues. Transfer learning is proposed in this study as a solution to this challenge.

Transfer learning leverages knowledge from a pre-trained model to learn new tasks in a target domain. This approach is particularly effective when target data are scarce, leading to extensive research in deep transfer learning methods for skin cancer diagnosis and classification. Pre-trained models in transfer learning are models that have been trained on large datasets and can be fine-tuned for specific tasks. These models are often used as starting points to speed up training and enhance deep learning model performance.

The Xception model stands out among pre-trained models due to its exceptional performance. Developed by Chollet [20], Xception is a depth-wise separable convolutional neural network architecture trained on the ImageNet database, which contains more than one million images. This extensive training has enabled the model to learn comprehensive features from diverse image types. Notably, Xception has consistently outperformed other architectures like VGG-16, ResNet, and Inception V3 in various traditional classification tasks. Studies have demonstrated Xception’s effectiveness in skin cancer detection through the analysis of dermoscopic images. Its ability to distinguish subtle differences in skin lesions makes it a powerful tool for early diagnosis [21,22]. The application of Xception in clinical settings can assist dermatologists in performing more accurate analyses, ultimately leading to improved patient outcomes and timely treatment.

Deep attention mechanisms (AMs) have gained significant attention in medical image classification [23,24] due to their promising results when combined with deep learning (DL) algorithms. Unlike traditional approaches that treat all image regions equally, AMs focus on specific areas of interest, potentially improving classification performance. This study focuses on three well-known types of attention mechanisms: soft attention, hard attention, and self-attention, each with a distinct approach to focusing on relevant data.

Soft attention provides a differentiable mechanism that allows models to smoothly focus on different input parts with varying emphasis, avoiding abrupt decisions. Hard attention makes binary decisions about which input parts to consider, establishing a clear relationship between the mechanism’s input and the neural network’s target. Self-attention, in contrast, focuses on modeling relationships within a single context, efficiently handling long-distance interactions. This is particularly powerful in tasks where the context is crucial and the relationship between different elements is key to understanding the whole [25].

Despite advances in skin cancer detection, further research is needed to explore techniques that enhance accuracy. No previous research has explored attention mechanisms with pre-trained Xception transfer learning for binary skin cancer classification. This study investigates Xception-based deep transfer learning, both with and without attention mechanisms, in detecting benign and malignant skin lesions. The Xception model was chosen due to its demonstrated effectiveness and strong performance, as demonstrated in prior research [26,27]. This model could help in the early detection of skin cancer, which, in turn, could enhance the chances of successful treatment.

The major contributions of this study are presented below.

Proposal of a novel model based on the Xception architecture that incorporates various AMs for binary classification of skin lesions as benign or malignant.A thorough investigation of how different AMs impact the Xception model’s performance.Comparison of the proposed models with recent state-of-the-art skin cancer detection methods in binary classification, using the same dataset.

This study is divided into the following sections. Section 2 presents the related works; the materials and methods are discussed in Section 3; the experimental results and discussion are presented in Section 4; and the study draws conclusions in Section 5.

## 2. Related Work

In recent years, numerous studies have employed deep-learning-based approaches to diagnose and classify skin cancer. These approaches have demonstrated improved performance compared to traditional machine learning methods [27,28]. The most recent literature uses convolutional neural networks (CNNs) and has shown competitive performance in diagnosing skin lesions, as demonstrated in studies [29,30,31,32,33], to name several examples. One recent study [30] used a CNN to classify seven skin cancer types from the HAM10000 dataset. Using the Enhanced Super-Resolution Generative Adversarial Network (ESRGAN) for image enhancement, the model achieved accuracies of 98.77%, 98.36%, and 98.89% for protocols I, II, and III, respectively.

A CNN was also proposed in [34] for skin lesion detection and classification, using systematic meta-heuristic optimization and CNN-based image detection techniques. The study compared various Keras classifiers, including MobileNet, on the HAM10000 and ISIC-2017 datasets. The proposed system outperformed the others, achieving 78% accuracy.

Deep convolutional networks have excelled in image segmentation and localization. Nawaz et al. [35] proposed a novel approach to locating and segmenting melanoma cells using faster region-based convolutional neural networks (RCNNs) with fuzzy k-means clustering. They applied this method to three datasets: ISBI-2016, ISIC-2017, and PH2. Their approach outperformed existing methods, achieving average accuracies of 95.40%, 93.1%, and 95.6% on the respective datasets.

Another study [36] detected melanoma using a novel lightweight convolutional neural network (LWCNN). The study utilized the HAM10000 dataset, with images labeled as either melanoma or non-melanoma. The proposed LWCNN model outperformed pre-trained models like GoogLeNet, ResNet-18, and MobileNet-v2. The model achieved 91.05% accuracy, with only 22.54 min of total processing time. On the same dataset, Renith and Senthilselvi [37] proposed a novel approach for classifying skin lesions as benign or malignant. Their model combined an improved AdaBoost algorithm with Aphid-Ant Mutualism optimization, built on the AlexNet architecture. This approach outperformed existing methods, achieving 95.7% accuracy, 95% specificity, 94.8% sensitivity, 95.4% precision, and a 95% F-measure.

Recent research has focused on exploring transfer learning approaches. In [38], a MobileNetV2-based deep transfer learning model was introduced for melanoma classification using 33,126 images extracted from the SIIM-ISIC 2020 dataset. The model achieved 98.2% accuracy, outperforming other techniques in terms of both accuracy and computational efficiency.

The authors of a different study [39] introduced SCDNet, a novel method based on VGG16 and CNN architectures, to classify 25,331 images from ISIC 2019 into four skin cancer categories. This model achieved 96.91% accuracy, surpassing the performance of ResNet50 at 95.21%, AlexNet at 93.14%, VGG19 at 94.25%, and Inception V3 at 92.54%.

The pre-trained ResNet and InceptionV3 models were used in [40] to extract features from 14,033 dermoscopic images from the HAM 10000, ISBI 2016, and ISBI 2017 datasets. After applying data augmentation techniques, a CNN was used for classification. The method achieved accuracies of 89.30%, 97.0%, and 94.89% on the datasets.

Imran et al. [41] developed an ensemble CNN model that combined VGG, CapsNet, and ResNet to detect cancerous and non-cancerous conditions. They used 25,000 images from the ISIC dataset. The results demonstrated that this ensemble approach of pre-trained models achieved 93.5% accuracy, with a training time of 106 s, outperforming the individual models across various performance metrics. 

Accurate detection and diagnosis of skin lesions can be challenging due to the similarity in appearance of various types of lesions, such as melanoma and nevi, especially in color images. Ashraf et al. proposed an automatic classification approach using a pre-trained AlexNet CNN model with a region of interest (ROI) for accurately extracting discriminative features of melanoma [42]. The model, trained on 3738 images from DermIS and DermQuest with extensive augmentation, achieved 97.9% and 97.4% accuracy on the datasets.

There has been growing interest in the comparative analysis of different transfer learning approaches. One study [43] evaluated six networks (VGG19, InceptionV3, InceptionResNetV2, ResNet50, Xception, and MobileNet) on the HAM10000 skin cancer dataset. Xception outperformed the others, achieving 90.48% accuracy, 89.57% recall, 88.76% precision, and an 89.02% F-measure.

Transfer learning models have encouraged research into their effectiveness for classifying skin cancer images in both binary and multiple categories. In [44], the authors evaluated modified EfficientNet V2-M and B4 models for both multiclass and binary classification of skin lesions, using 58,032 dermoscopic images from the ISIC datasets. EfficientNetV2-M outperformed the others in both tasks. For multiclass classification, it achieved accuracies of 97.62%, 95.49%, and 94.8% on the ISIC 2020, 2019, and HAM datasets, respectively. In binary classification, it reached 99.23%, 97.06%, and 95.95% accuracy on the same datasets.

Another study [45] performed binary (cancer vs. non-cancerous) and multiclass (six lesion types) classification on 2298 dermoscopy images from the PAD-UFES-20 dataset, using a CNN-based transfer learning model. The pre-trained CNN model improved the accuracy rates by over 20% compared to conventional models. The proposed model achieved a mean accuracy and an F1 score of 0.86 for both classification types.

Many studies have used pre-trained models on the HAM10000 dataset for binary melanoma classification. These studies typically focus on melanoma cases, which comprise 1113 instances in the dataset, while classifying the remaining skin lesion types as non-melanoma. One such study [46] tested four pre-trained models (ResNet50, InceptionV3, Xception, and VGG16) with three segmentation techniques (SegNet, BCDU-Net, and U-Net). The results showed that Xception performed best, achieving accuracies of 95.2% and 95.2% with BCDU-Net and SegNet, respectively. The study in [47] used the augmented HAM and PH2 datasets, totaling 18,004 images, to classify melanoma. Combining handcrafted features, EfficientNet-B0 and hair removal techniques, they achieved 94.9% and 98% accuracy on the HAM and PH2 datasets, respectively. Another study [48] compared a non-pre-trained CNN with three pre-trained models (MobileNetV2, EfficientNetV2, and DenseNet121) for melanoma classification on the HAM dataset. The pre-trained models achieved better overall accuracies of 93.77%, 93.60%, and 93.34%, respectively, compared to the CNN, which achieved 91.35%.

Previous studies focused on exploring deep learning and transfer learning techniques, treating all the image patches equally. A shift in the research introduced deep attention mechanisms to highlight regions of interest and extract optimal features, potentially enhancing the skin cancer detection accuracy. In [49], the authors proposed a soft-attention-based convolutional neural network (SAB-CNN) for classifying HAM dataset images. SMOTE was used to address the dataset imbalance. The model achieved 95.94% accuracy, a 95.30% Matthews correlation coefficient, and a 95.97% balanced accuracy score. This study highlighted the importance of attention mechanisms and data balancing in improving deep neural network performance.

The use of AMs has encouraged research investigating the effectiveness of CNNs with AMs [24,50,51]. One study [24] examined soft attention’s impact on five deep neural networks (ResNet34, ResNet50, Inception ResNet v2, DenseNet201, and VGG16) for skin cancer image classification. Soft attention aims to enhance crucial elements while reducing noise. The results showed a 4.7% improvement over the baseline, achieving 93.7% precision on the HAM10000 dataset. On the ISIC-2017 dataset, soft attention coupling improved the sensitivity by 3.8%, reaching 91.6% compared to other approaches.

Another study [23] proposed a dual-track deep learning model for skin cancer classification. The first track used a modified DenseNet-169 with a coordinate attention module (CoAM) for local features, while the second employed a custom CNN with a feature pyramid and global context networks for multiscale and global features. By combining these features, the model achieved 93.2% accuracy, 95.3% precision, 91.4% recall, and a 93.3% F1 score on the HAM10000 dataset.

Table 1 provides a concise overview of the related works, revealing that few studies have combined attention mechanisms with pre-trained models for binary skin cancer classification. Notably, no previous research has explored the use of deep attention mechanisms with the pre-trained Xception model for diagnosing skin cancer in a binary classification context. The studies reviewed in the previous section demonstrate that integrating AMs has shown promising accuracy in extracting spatial information and highlighting regions of interest in images.

This encourages further investigation into the effects of integrating Xception-based deep transfer learning with AMs for detecting benign and malignant skin lesions. Our contribution in this work is to explore and compare the use of different types of deep attention mechanisms with Xception in the detection of skin cancer. Based on a review of related works, this approach has not previously been employed.

## 3. Materials and Methods

As previously stated, this study investigates the impact of integrating Xception deep transfer learning methods with different attention mechanisms (SL, SF, and HD) for detecting skin cancer in dermoscopy images. Figure 1 illustrates the architecture of the proposed model.

In the following sections, the implementation of the main components of our proposed models is described. This includes the dataset description, data augmentation, data preprocessing, Xception-based models for feature extraction, deep attention integration, image classification, and model evaluation. The specifications related to each model are identified, both with and without attention mechanisms (AMs), highlighting any differences between them.

### 3.1. Dataset

In this study, the HAM10000 (“Human Against Machine”) dataset [3,53], a collection of pigmented skin lesion images publicly available on the ISIC archive, was used [54]. The dataset comprises 10,015 images representing seven types of pigmented skin lesions. These types include actinic keratosis (AKIEC), basal cell carcinoma (BCC), benign keratosis (BKL), dermatofibroma (DF), melanocytic nevi (NV), melanoma (MEL), and vascular skin lesions (VASC). More than 50% of the lesions were verified through histopathology, while the remaining cases were confirmed via follow-up examination, expert consensus, or in vivo confocal microscopy. All the images are in color, with dimensions of 450 × 600 pixels. Figure 2 illustrates examples of these seven lesion types, while Figure 3 displays their class distribution. The *x*-axis represents the lesion types, and the *y*-axis shows their corresponding counts.

As shown in Figure 4, the dataset is well balanced in terms of the sex distribution, with 51.1% of the images belonging to males and 48.9% to females. The age distribution exhibits a bimodal pattern, with peaks at around 35–50 years old and 60–75 years old. This pattern indicates that the majority of patients in the dataset are either middle-aged or elderly, which is consistent with the higher prevalence of skin cancer in older adults.

As the HAM10000 dataset is multi-class, this study focused on binary classification. The seven classes were classified into either malignant (cancerous) or benign (normal) groups. MEL and BCC were grouped as cancer, while DF, BKL, NV, VASC, and AKIEC were identified as normal. The dataset thus comprised two binary classes: cancer and normal, as illustrated in Table 2.

The normal class comprised 80% of the entire image dataset, leading to an imbalanced database, which significantly impacted the training process. The cancer class represented only 19.56% of the images, with the AKIEC labels constituting less than 3% of the total. To ensure a balanced dataset, data augmentation techniques were implemented.

### 3.2. Data Augmentation

Various data augmentation techniques were applied to the cancer class, including rotation, brightness adjustment and flipping, using the real-time image data generator function from the Keras library in Python 2.15.0. These techniques increased the sample size of the cancer class and enhanced the diversity of the training data.

The original images were rotated by up to 40°, applying a random rotation angle between −40° and 40°. The brightness of the images was rotated to between 1.0 and 1.3 times the original brightness, simulating different lighting conditions. We also flipped the images randomly, both vertically and horizontally. The parameters and their selected values are provided in Table 3.

After augmentation, three augmented versions were generated for each original cancer image, increasing the size of the cancer class from 1954 to 7816 images. This brought the cancer class size much closer to the normal class size, resulting in a total dataset of 15,877 dermoscopy images. The statistics in Figure 5 illustrate the differences in the dataset before and after augmentation.

### 3.3. Data Preprocessing

After augmenting the images, preprocessing and preparation procedures were applied to the dermatoscopic images from the HAM10000 dataset. These procedures encompassed resizing, normalization, and data shuffling. The dermatoscopic images were resized from 450 × 600 pixels to 299 × 299 pixels to match the Xception model’s default input size. Pixel values from the 0–255 range to the 0–1 range were then standardized, which is optimal for neural network models. Finally, to prevent bias during training, data shuffling was applied, ensuring randomness in batch selection and preventing the model from learning patterns based on the order of the data.

### 3.4. Feature Extraction with Pre-Trained Xception Models

The Xception base model serves as a powerful feature extractor due to its pre-training on the extensive ImageNet dataset. When loading the Xception model, the “include top” parameter was set to “false”. This meant that the fully connected top layers originally designed for the ImageNet classification task were not loaded. Instead, these layers were replaced by custom fine-tuning layers suited to the specific classification task at hand. This approach leveraged pre-learned knowledge, provided flexibility for customization, improved accuracy, enhanced computational efficiency, and substantially reduced the number of unnecessary parameters.

Following the base Xception model, a GlobalAveragePooling2D layer was added to reduce the spatial dimensions by extracting global features from the feature maps generated by the Xception base model. This was followed by a dropout layer to prevent overfitting, enhancing the model’s generalization capabilities.

### 3.5. Deep Attention Integration

An attention layer was integrated into the Xception architecture, which was one of three mechanisms: hard, soft, or self-attention. Figure 6 shows an illustration of the Xception model with AMs.

Each of these mechanisms was integrated separately into the overall architecture, utilizing the same preceding and subsequent layers and procedures. Each attention mechanism (AM) layer has its own method for analyzing the features extracted from the base Xception model. These methods determined which parts of the image were most important for the classification task, as follows:

SL layer: This layer transformed the input into query (Q), key (K), and value (V) vectors through linear transformations. The attention scores were computed as the dot product of the query with all the keys divided by $\sqrt{d_{k}}$. These scores were then normalized using Softmax to obtain the attention weights for the values [55]. In this project, self-attention was internally implemented using Keras’s built-in attention layer [56], following this equation:(1)$$SelfAttention\left(Q,K,V\right)=softmax\left(\frac{QK^{T}}{\sqrt{d_{k}}}\right)V$$SF layer: This layer discredited irrelevant areas of the image by multiplying the corresponding feature maps with low weights. The low attention areas had weights closer to 0, allowing the model to focus on the most relevant information, which enhanced its performance [24]. A dense layer was used with Softmax activation to compute the attention weights $\alpha_{i}$ for each feature $x_{i}$, where Softmax ensured that these weights sum to 1, as shown in the following equation [57]:(2)$$\alpha_{i}=\frac{exp(w_{i}\cdot x_{i})}{\sum_{j=1}^{n}exp(w_{j}\cdot x_{j})}\;for\;i=1,2,\ldots ,n$$These attention weights were then applied to the feature map x using a dot product operation:(3)$$y=\sum_{i=1}^{n}\alpha_{i}x_{i}$$HD layer: This layer compelled the model to focus exclusively on crucial elements, disregarding all others. The weight assigned was either 0 or 1 for each input component. This applied a binary mask to the attention scores between the queries Q and the keys K. The mechanism assigned a value of 1 to the top k highest-scoring elements (selected by TopK), and 0 to the rest [58]. This forced the model to focus only on the most important elements, disregarding others, without involving gradients in the selection process. The process is represented by the following equation:(4)$$A_{hard}\left(Q,K\right)=1\left[score\left(Q,K\right)\in TopK\left(score\left(Q,K\right),k\right)\right]$$Based on the attention mechanism’s type and analysis, a weighted feature map was created, assigning higher weights to the more relevant features and lower weights to the less important ones.

### 3.6. Image Classification

This is the last layer in the Xception model’s architecture. At this stage, the input images were classified into two main categories—normal and cancer—using the following process.

#### 3.6.1. Dense (Fully Connected) Layer

The output from the AM’s layers was flattened and fed into the dense layer. This process combined all the information gathered from the network’s previous layers to classify the input image.

#### 3.6.2. Sigmoid Layer

A sigmoid function was utilized to transform the output of the fully connected layer into a binary (0 or 1), which was interpreted as the classification probability.

#### 3.6.3. Classification Layer

The dense layer with a sigmoid activation function served as the final classification layer. In the case of the integrated AMs, it took the output of the AM layers as input and generated the final classification predictions into one of two classes (normal or cancer). To mitigate overfitting, L2 regularization techniques were applied to the weights of this dense layer. In the original Xception-based model without AMs, the dense layer with sigmoid activation took the learned features from the Xception base model and the global average pooling layer to make the final class predictions. L2 regularization was not used for this model as it was unnecessary.

### 3.7. Model Evaluation

After the training process, the proposed models were tested on the testing dataset. The architecture’s performance was evaluated using the accuracy, F1 score, precision, and recall. These performance metrics are explained in detail below, along with their definitions and equations. In these equations, TP stands for true positives, TN for true negatives, FN for false negatives, and FP for false positives.

#### 3.7.1. Classification Accuracy

The model’s ability to correctly classify samples compared to the total number of samples in the evaluation dataset is known as accuracy. It is calculated as follows:(5)$$Accuracy=\frac{\#\;correctly\;classified\;samples}{\#\;all\;samples}=\frac{TP+TN}{TP+FP+TN+FN}$$

#### 3.7.2. Recall

It is also referred to as sensitivity or the true positive rate (TPR), indicating the rate of correctly classified positive samples. This metric is considered a crucial factor in medical research, as the aim is to miss as few positive cases as possible, leading to high recall [59].
(6)$$REC=\frac{\#\;true\;positive\;samples}{\#\;samples\;classified\;positive}=\frac{TP}{TP+FN}$$

#### 3.7.3. Precision

This represents the proportion of retrieved samples that are pertinent and is computed as the ratio of correctly classified samples to all the samples assigned to a specific class.
(7)$$PREC=\frac{\#\;samples\;correctly\;classified}{\#\;samples\;assigned\;to\;class}=\frac{TP}{TP+FP}$$

#### 3.7.4. F1 Score

As a widely used metric in binary and multi-class classification, the F1 score combines precision and recall through their harmonic mean. It balances these metrics, making it especially valuable for imbalanced datasets.
(8)$$F1=2\times \frac{precision\times recall}{precision+recall}=\frac{2\times TP}{2\times TP+FP+FN}$$

#### 3.7.5. False Alarm Rate (FAR)

Incorrectly classifying negative instances as positive occurs at a rate measured by the false positive rate. This metric quantifies how often a model raises erroneous alerts for cases that are actually negative.
(9)$$FAR=\frac{Incorrectly\;classified\;actual\;negatives}{All\;actual\;positives}=\frac{FP}{FP+TN}$$

#### 3.7.6. Cohen’s Kappa

This quantitative measure assesses the level of agreement between two raters evaluating the same subject, while accounting for the possibility of chance agreement. This metric is widely adopted across various fields, including statistics, psychology, biology, and medicine.
(10)$$k=\frac{P_{A}-P_{E}}{1-P_{E}}$$

#### 3.7.7. AUC Score and ROC Curve

The receiver operating characteristic (ROC) curve graphically illustrates the relationship between the false positive rate and the true positive rate of a classifier. The probability curve’s area under the curve (AUC) indicates how well the model distinguishes between classes. A higher AUC value reflects better class separation by the classifier.

## 4. Results and Discussion

In this section, the results and findings of the Xception architecture incorporating different AMs are discussed, beginning with an explanation of the experimental settings and then analyzing the results of the four developed models, comparing them to those of previous studies.

### 4.1. Experimental Settings

The proposed models were trained for skin cancer diagnosis on dermatoscopic images from the HAM10000 dataset. Augmentation techniques were applied to expand the sample size and enhance the diversity of the training data. The dataset was split into 20% for testing and 80% for training. A 10-fold cross-validation technique was used for training and testing the models. Table 4 provides a detailed breakdown of the dataset split. To prevent bias, the order of the images was shuffled randomly using a data shuffling technique.

All four proposed models were trained using the Adam optimizer with a learning rate of 0.001, a batch size of 32, and a total of 50 epochs across all the folds. The binary cross-entropy loss function was used, with a default probability threshold of 0.5. The activation function in the model with AMs includes both sigmoid and Softmax, while the model without AMs uses only sigmoid. Table 5 provides an overview of the optimization hyperparameters used in the experiments.

Early stopping was applied, with a patience of five epochs, to prevent overfitting. Additionally, an error-handling technique was used that ensured robust training and maintained optimization integrity by handling errors. The best model weights were saved, based on decreased validation loss and higher accuracy.

### 4.2. Classification Results

Four experiments were conducted to evaluate the performance of the Xception model, both with and without three attention mechanisms. These experiments compared the base Xception model to versions incorporating different types of AMs, resulting in four distinct models: Xception (base), Xception-SL, Xception-SF, and Xception-HD. The performance of the proposed models was evaluated by measuring the accuracy, recall, precision, F1 score, Cohen’s kappa, and AUC scores for each model. 

Table 6 shows the results of the four models.

The results of the experiments revealed a significant effect of the AMs on the Xception model’s performance. Incorporating AMs into the Xception model led to improvements across all the metrics. In contrast, the base Xception alone exhibited the lowest performance on all the metrics. Nevertheless, the results were good.

The results also showed a convergence between the three AM models with Xception integration. Incorporating self-attention (Xception-SL) into the Xception architecture yielded the highest performance across all the metrics. This suggests that the self-attention mechanism significantly improved the Xception architecture by capturing the relationships between distant elements in a sequence.

The Xception-SF model demonstrated the second-best performance across all the metrics, except for precision, when compared to Xception-HD. Both models yielded similar and promising results. Notably, Xception-SF achieved a recall of 95.28%, almost matching that of Xception-SL.

The results strongly suggest that incorporating deep attention mechanisms improved Xception’s performance, particularly in terms of recall—a critical metric in medical applications. All the AM-enhanced Xception models achieved promising results. These recall performances indicate that each variant showed promise in effectively identifying skin cancer cases.

After examining the performance of the models through the recall, accuracy, precision and F1 score, their agreement was evaluated using Cohen’s kappa, which provided insights into the agreement between the models’ predictions and the true labels. 

Table 6 presents the aggregate Cohen’s kappa scores obtained from 10-fold cross-validation. The Xception models, both with and without the three AMs, demonstrated strong performance for this metric. All the models achieved scores between 0.821 and 0.882, indicating substantial agreement between their predictions and the ground truth.

Having discussed the performance of the models using Cohen’s kappa, their performance was evaluated, as reflected in the AUC scores. Figure 7 illustrates the ROC curves and AUC results of the four models. The AUC score reflects each model’s ability to differentiate between classes—in this case, normal and cancer. The Xception models with the three AMs achieved convergence in their results, which were all 0.98, while the Xception performed slightly lower, with a score of 0.97.

Lastly, the confusion matrix results present a detailed view of the models’ classification performance. Figure 8 shows the confusion matrices for the four Xception models (the base Xception, Xception-SL, Xception-SF, and Xception-HD).

As shown in Figure 8, the Xception models with the three AMs exhibited similar performance patterns. The Xception-SL model demonstrated the best performance, correctly classifying 1539 normal pigmented skin images and 1449 cancerous images, with 114 false positives and 73 false negatives. The Xception-SF model accurately classified 1536 normal pigmented skin images and 1426 cancerous images, with 137 false positives and 76 false negatives. The Xception-HD model exhibited a slightly different pattern, correctly classifying 1515 normal pigmented skin images and 1437 cancerous images, but with higher numbers of 126 false positives and 97 false negatives.

In contrast, the base Xception model correctly identified 1478 normal pigmented skin images and 1413 cancerous images, with the highest numbers of 150 false positives and 134 false negatives. These results highlight the varying impacts of different AMs on the models’ classification accuracy and error distribution.

From the confusion matrix, the false alarm rate was determined for each model. The Xception-SL model achieved the best results, with the highest classification rate of 94.11% and the lowest false alarm rate among all the models at 6.90%. The Xception-HD and Xception-SF models demonstrated similar performance. The Xception-HD model achieved a classification rate of 92.97%, with a false alarm rate of 7.68%, while the Xception-SF model slightly outperformed it with a classification rate of 93.29% but a slightly higher false alarm rate of 8.19%. The Xception alone had the highest false alarm rate of 9.21%, which negatively impacted its overall classification rate of 91.05%. These results highlight the significant impact of the different AMs on the models’ ability to accurately classify skin lesions while minimizing false positives.

Our extensive performance evaluation and analysis indicates that the Xception model, incorporating three attention mechanisms (self, hard, and soft), consistently outperformed the standard Xception model across all the metrics. These findings strongly suggest that integrating attention mechanisms into the Xception architecture effectively enhances the detection of pigmented skin lesions in dermatoscopic images. This highlights the potential of attention mechanisms to improve performance in relation to complex medical imaging tasks, particularly in oncological applications.

Despite the promising results achieved by our proposed models, several factors affected the models’ ability to achieve higher performances. One factor is the limited size and diversity of the HAM1000 dataset, which led to overfitting. While the Xception-based models with/without AMs performed exceptionally well on the training set, their test set performance was lower than expected, indicating generalization challenges. Although image augmentation and anti-overfitting techniques, including L2 regularization, early stopping, and dropout layers, were employed, overfitting remains a concern. Future research could explore generative AI techniques, particularly generative adversarial networks (GANs), to create diverse synthetic images. To mitigate overfitting and enhance model robustness, additional strategies could be investigated, such as implementing weight decay and expanding the training dataset. 

Data quality is another crucial factor that may have affected the performance of the models. The HAM10000 dataset suffered from image noise caused by varying lighting conditions, different device types used for capture, inconsistent image resolution, and the variable clarity of the lesion boundaries. This might have introduced inconsistencies into the models’ learning process, potentially hindering their ability to identify and learn relevant features. Continuous efforts in noise-filtering techniques could improve the image quality and readability, potentially enhancing the models’ classification accuracy.

The high computational resources required for training and evaluating the four developed models should also be considered, especially as the complexity of the fine-tuning architecture increased. Despite these challenges, the proposed models showed promising enhancements of skin cancer detection and classification.

### 4.3. Comparison with Other Models

Diverse transfer learning approaches applied to the HAM10000 dataset were investigated. Two recent and closely related studies were identified for comparison with our proposed method. One study was published in 2023 and the other in 2024, after our experiments were completed.

These two studies were selected for comparison because they share several key characteristics with our research. Both studies divided the HAM10000 dataset into malignant and benign categories, maintained a similar distribution of samples, and employed transfer learning models. One of the studies also incorporated deep attention mechanisms. Table 7 (below) compares the performance of our proposed models on the HAM10000 dataset with the results from the two studies. Both studies reported the accuracy and provided additional metrics, including the recall, precision, and F1 score.

The study in [44] reported the highest accuracy of 95.95%, using modified versions of EfficientNet V2-M and EfficientNet-B4 for classifying malignant and benign skin lesions. While this accuracy was slightly higher than that of our best-performing Xception-SL model (94.11%), our proposed models outperformed theirs on all the other key metrics, including the recall, precision, and F1 score. Specifically, our Xception-SL model achieved recall, precision, and F1 score values of 95.47%, 93.10%, and 94.27%, respectively, while our Xception-SF model achieved 95.28%, 91.81%, and 93.51%. Additionally, our Xception models incorporating attention mechanisms recorded higher AUC scores than the EfficientNet models, with an improvement of more than 0.03.

The other study in [23] demonstrated that integrating a modified DenseNet-169 network with a coordinate attention mechanism (CoAM) and a customized CNN improves the precise localization and modeling of long-range dependencies in dermoscopic images. This approach achieved the highest precision of 95.3%, an accuracy of 93.2%, a recall of 91.4%, and an F1 score of 93.3%.

Our proposed approach of integrating three different AMs (SF, HD, and SL) into the Xception architecture significantly enhanced the network performance. This integration selectively focused on the most relevant areas of skin lesion images, improving accuracy and capturing long-range dependencies.

Our results were competitive in terms of both the accuracy and F1 score. The Xception-SL and Xception-SF models achieved higher accuracy than the approach in [23], with Xception-SL being the most accurate at 94.11%, followed by Xception-SF at 93.29%. Both models also outperformed [23] in terms of the F1 score, with an improvement of more than 0.48 points. Although the precision of our four developed models was slightly lower, the recall is often a more critical measure than the precision in medical applications, as minimizing false negatives is essential to ensure that as few actual cases as possible are missed. All our proposed models (Xception-SL, Xception-SF, Xception-HD, and Xception) outperformed the approach of these studies in terms of the recall, with scores of 95.47%, 95.28%, 93.98%, and 91.68%, respectively.

In summary, our proposed models demonstrate promising advancements compared to recent studies in classifying and detecting skin cancer. Notably, both the self-attention and soft attention models outperformed the previous studies in the recall metric, a critical measure for medical investigations.

These results show that integrating different AMs into the Xception architecture improves the classification of malignant and benign skin lesions, potentially enhancing medical diagnostics and patient care.

## 5. Conclusions

In this study, a novel model based on the Xception architecture was proposed, incorporating three attentional mechanisms (self, hard, and soft) to classify skin cancer as benign or malignant. The impact of these AMs on model performance was thoroughly investigated. The results demonstrate that integrating AMs into the Xception architecture effectively enhances its performance. The accuracy of Xception alone was 91.05%. With the AMs, the accuracy increased to 94.11% with self-attention, 93.29% with soft attention, and 92.97% with hard attention. Notably, both the self-attention and soft attention models outperformed previous studies on the recall metric, which is crucial for medical investigations. To the best of our knowledge, this is the first study to investigate the impact of attention mechanisms in Xception-based deep transfer learning for binary skin cancer classification. The findings suggest that attention mechanisms can enhance pre-trained models, with potential applications in aiding dermatologists in early skin cancer diagnosis, potentially improving treatment outcomes and survival rates. 

A limitation of our study was the limited size and diversity of the HAM10000 dataset, which led to overfitting. Although image augmentation and anti-overfitting techniques were employed, overfitting remains a concern. Additionally, image noise may have impeded the ability of the models to learn relevant features effectively, reducing their overall accuracy and performance. Future work will focus on experimenting with larger combined datasets and using GANs to synthesize realistic skin lesion images, implementing noise filtering techniques and exploring various attention mechanisms to improve the model performance. Transfer learning approaches will be evaluated using EfficientNet and ResNet, along with ensemble methods.

## Figures and Tables

**Figure 1 diagnostics-15-00099-f001:**
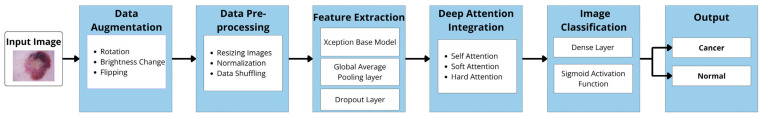
Model architecture.

**Figure 2 diagnostics-15-00099-f002:**
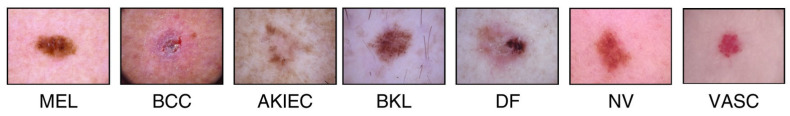
HAM10000 dataset.

**Figure 3 diagnostics-15-00099-f003:**
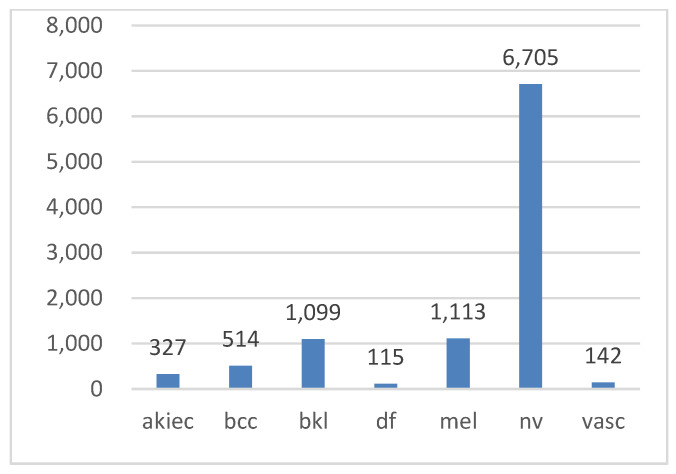
Class distribution in the Ham10000 dataset.

**Figure 4 diagnostics-15-00099-f004:**
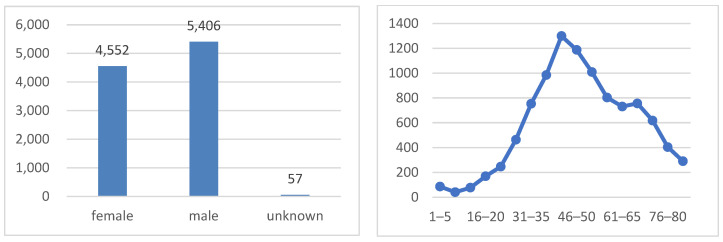
Sex class distribution (on the **left** side) and age distribution (on the **right** side).

**Figure 5 diagnostics-15-00099-f005:**
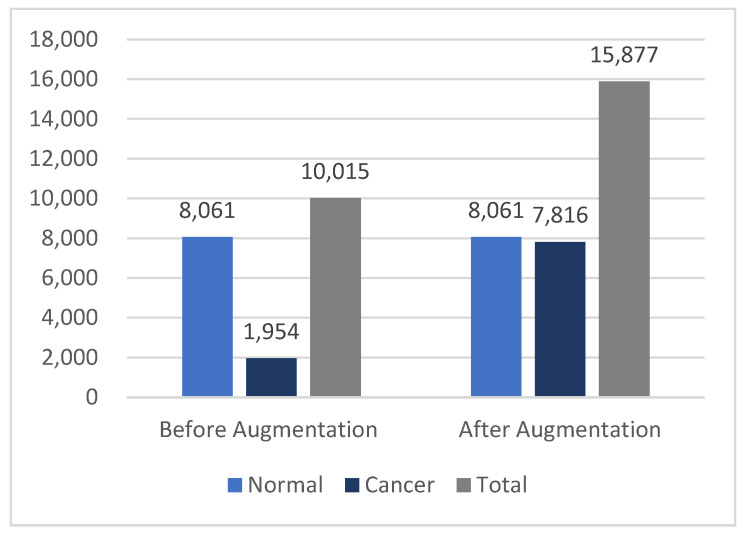
Differences in the HAM10000 dataset size before and after data augmentation.

**Figure 6 diagnostics-15-00099-f006:**
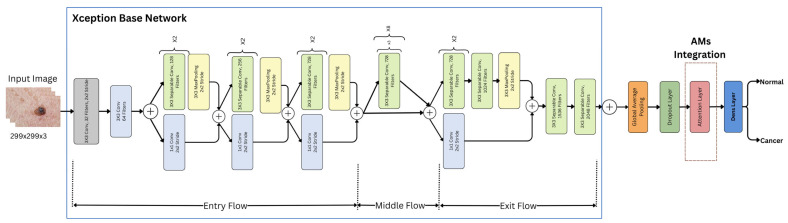
Xception model with AMs.

**Figure 7 diagnostics-15-00099-f007:**
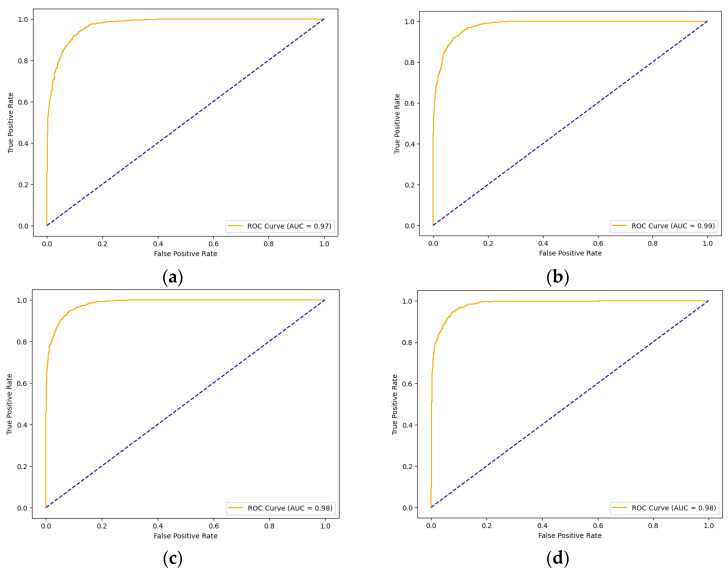
ROC curves of the models for skin cancer: the ROC curve of the Xception model (**a**); the ROC curve of the Xception-SL model (**b**); the ROC curve of the Xception-HD model (**c**); and the ROC curve of the Xception-SF model (**d**).

**Figure 8 diagnostics-15-00099-f008:**
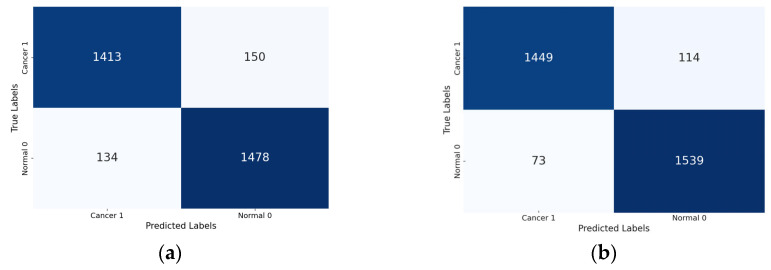
Confusion matrices for the Xception models with/without AMs: Xception model (**a**); Xception-SL model (**b**); Xception-HD model (**c**); and Xception-SF model (**d**).

**Table 1 diagnostics-15-00099-t001:** Skin cancer detection and classification studies.

Ref	Approaches	Dataset	Classification Type	Evaluation Metrics			
				Precision	Recall	Accuracy	F1 Score
[34]	CNN	HAM1-ISIC 2017	Multi-class	NA	NA	78%	NA
[30]	CNN	HAM	Multi-class	NA	NA	98.89%	NA
[35]	RCNN-FKM	ISIC-2016 ISIC-2017 PH2	Binary	NA	97.2%	96.1%	NA
[37]	AdaBoost + IAB-AAM + AlexNet	HAM	Binary	95.4%	94.8%	95.7%	95%
[36]	LWCNN	HAM	Binary	NA	NA	91.05%	NA
[39]	CNN-VGG16	ISIC 2019	Multi-class	92.19%	92.18%	96.91%,	92.18%
[42]	Pre-trained CNN model AlexNet + ROI	DermIS DermQuest	Binary	NA	NA	97.9%	NA
[38]	MobileNetV2	ISIC-2020	Binary	98.3%	98.1%	98.20%	98.1%
[45]	Pre-trained CNN	PAD-UFES-20	Binary/multi-class	B = 88% M = 90%	B = 81% M = 83%	B = 86% M = NA	B = NA M = 86%
[52]	Modified VGG16 architecture	Kaggle	Binary	NA	NA	89.09%	93.0%
[41]	CNN-VGGNet, CapsNet, and ResNet	ISIC	Multi-class	94%	NA	93.5%	92.0%
[40]	CNN-ResNet, InceptionV3	ISBI 2016 ISBI 2017 Ham	Multi-class	95.30%	NA	95.89%	94.90%
[47]	HC + ResNet50V2 and EfficientNet	HAM and PH2	Binary	92.8%	97.5%	98%	95%
[44]	EfficientNet V2-M and EfficientNet-B4	ISIC 2020, ISIC 2019 HAM	Multi-class/Binary	B = 96% M = 96%	B = 95% M = 95%	B = 97.06% M = 95%	B = 95% M = 95%
[43]	Six transfer learning networks	HAM	Multi-class	88.76%	89.57%	90.48%,	89.02%
[46]	Four pre-trained + image segmentation	Ham	Binary	NA	94.16%	96.10%	96.02%
[48]	MobileNetV2, EfficientNetV2 + DenseNet121 + CNN	Ham	Binary	93.77%	89.78%	93.77%	93.51%
[49]	CNNs + soft attention	Ham	Multi-class	NA	NA	95.94%	NA
[24]	Six pre-trained models + soft attention	HAM and ISIC 2017	Multi-class	93.7%	NA	93.4%	NA
[23]	Densenet-169 with CoAM + customized CNN	HAM	Binary	95.3%	91.4%	93.2%	93.3%

**Table 2 diagnostics-15-00099-t002:** Image statistics for the Ham10000 dataset.

Cancer			Total	Normal				Total
MEL	BCC	AKIEC	1954 (19.56%)	DF	BKL	NV	VASC	8061 (80.49%)
1113	514	327		115	1099	6705	142	

**Table 3 diagnostics-15-00099-t003:** Parameters of the data augmentation.

Parameters	Values	Description
Rotation range	40	Randomly rotate images within a range of 40 degrees
Brightness range	[1.0, 1.3]	Adjust brightness 1.0–1.3 times original
Horizontal flip	True	Flipping the image horizontally
Vertical flip	True	Flipping the image vertically

**Table 4 diagnostics-15-00099-t004:** Data breakdown.

Dataset Size	Training Sets	Testing Sets
15,877	12,702	3175

**Table 5 diagnostics-15-00099-t005:** Hyperparameter settings: Xception-based models with and without AMs.

Parameter	With AMs	Without AMs
Epochs	50	50
Dropout	0.7	Not used
Shuffle	True	True
Activation function	Sigmoid/Softmax	Sigmoid
L2 regularization	0.001	Not used
Loss function	Binary cross-entropy	Binary cross-entropy
Probability threshold	0.5	0.5
Optimizer	Adam	Adam
Learning rate	0.001	0.001
Batch size	32	32

**Table 6 diagnostics-15-00099-t006:** Model classification results.

Models	Accuracy (%)	Recall (%)	Precision (%)	F1 Score (%)	AUC	Cohen’s Kappa
Xception (base)	91.05%	91.68%	90.78%	91.23%	0.972	0.821
Xception-SL	94.11%	95.47%	93.10%	94.27%	0.987	0.882
Xception-SF	93.29%	95.28%	91.81%	93.51%	0.983	0.865
Xception-HD	92.97%	93.98%	92.32%	93.14%	0.983	0.859

**Table 7 diagnostics-15-00099-t007:** Comparison with state-of-the-art models.

Ref/Year	Dataset Relabeling Method	Approach	Precision	Recall	Accuracy	F1 Score	AUC
[44] 2023	Benign = 8388 Malignant = 1627	EfficientNetV2-M and EfficientNet-B4	95.95%	94%	83%	88%	0.980
[23] 2024	Benign = 8061 Malignant = 1954	Modified DenseNet-169 with CoAM + customized CNN	93.2%	91.4%	95.3%	93.3%	-
Our Proposed Models	Normal = 8061 Cancer = 1954	Xception (base)	91.05%	91.68%	90.78%	91.23%	0.972
		Xception-SL	94.11%	95.47%	93.10%	94.27%	0.987
		Xception-SF	93.29%	95.28%	91.81%	93.51%	0.983
		Xception-HD	92.97%	93.98%	92.32%	93.14%	0.983

## Data Availability

The HAM10000 dataset is available at https://dataverse.harvard.edu/dataset.xhtml?persistentId=doi:10.7910/DVN/DBW86T (accessed on 26 December 2024).

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
