# Peer review of "Skin Cancer Detection Using Transfer Learning and Deep Attention Mechanisms"

_diagnostics, 2025, doi:10.3390/diagnostics15010099_

Round 1

Reviewer 1 Report

Comments and Suggestions for Authors

The article is devoted to skin cancer diagnostics using dermatoscope images and machine learning methods. The authors pay great attention to machine learning algorithms, but little to the data that these algorithms are used to process. The authors provide a very good overview of existing machine learning methods, their applicability is shown, and their comparison is presented in the form of a table. The article use known databases, but explanations for them are only briefly presented in the materials and methods section. It is written only about the fact that melanoma is examined by biopsy, and methods based on appearance are not accurate. It should at least mention the biophotonics methods that are in-vivo, have high accuracy, but at the same time are expensive.

As for the novelty, it is said that there is no work that would study the attention mechanisms with pre-trained Xception transfer learning for binary classification of skin cancer. However, a quick literature search found the following work [1]. In Work 1, an approach is presented with the Xception network with a soft attention-based framework.

Comments on the text:

The second paragraph requires revision and has duplicate information, for example

Skin cancer is the most prevalent, accounting for 75% of all cancer cases globally [3]. According to the World Health Organization (WHO), skin cancer accounts for one-third of global cancer cases.

Several risk factors may cause skin cell damage, including genetics, lighter natural skin color, older age, and certain medical conditions. It is unclear what medical conditions... can we expand it? Are tattoos one of these conditions?

Images 1 and 2,5,6,7 are of low quality and require improvement (it is impossible to see the words that are in the text and are key)

Image 3,4 does not have a caption for y - this is probably the number of examples with the presented class.

ROC-curves in Figure 6 should be combined into one graph and not presented on 4 different ones, since it is difficult to compare them in this form.

There is no point in specifying the confusion matrix (Figure 6) if all the data is already described in words in the text. Besides, metrics are presented in the table below.

Author Response

Thank you for your valuable feedback. Below, I have addressed each of your comments point by point.

Reviewer 2 Report

Comments and Suggestions for Authors

abstract

report briefly( in a sentece) the types of attention differences

line 18 - effectiveness based on what?

Intro

give with examples those types of attention and how they differentiate the results of the evaluation

also this approach is based on AI on dermoscopy so a little more on dermoscopy should be added

nice table

methods

excellent presentation with clear language of the concept and nice terminology

a question -comment For example in case of skin of colour patients with patterns in skin cancer(doi.org/10.3390/medicina60091386) and other diseases being inflammatory or infectious (pigmented elements, dermoscopy usually show dark backrounds-  dark or brown structures) that in case of certai types of attention would be misdiagnosed- what practical approach should be maintained in cases such as this- would be a additional limitation in the use

Author Response

(The authors gave the same response as above.)

Round 2

Reviewer 1 Report

Comments and Suggestions for Authors

Dear Authors,

Thank you for your efforts  to improve the manuscript.